SCIENCE FORUM

# Improving pandemic influenza risk assessment

**Abstract** Assessing the pandemic risk posed by specific non-human influenza A viruses is an important goal in public health research. As influenza virus genome sequencing becomes cheaper, faster, and more readily available, the ability to predict pandemic potential from sequence data could transform pandemic influenza risk assessment capabilities. However, the complexities of the relationships between virus genotype and phenotype make such predictions extremely difficult. The integration of experimental work, computational tool development, and analysis of evolutionary pathways, together with refinements to influenza surveillance, has the potential to transform our ability to assess the risks posed to humans by non-human influenza viruses and lead to improved pandemic preparedness and response.

COLIN A RUSSELL*, PETER M KASSON, RUBEN O DONIS, STEVEN RILEY, JOHN DUNBAR, ANDREW RAMBAUT, JASON ASHER, STEPHEN BURKE, C TODD DAVIS, REBECCA J GARTEN, SANDRASEGARAM GNANAKARAN, SIMON I HAY, SANDER HERFST, NICOLA S LEWIS, JAMES O LLOYD-SMITH, CATHERINE A MACKEN, SEBASTIAN MAURER-STROH, ELIZABETH NEUHAUS, COLIN R PARRISH, KIM M PEPIN, SAMUEL S SHEPARD, DAVID L SMITH, DAVID L SUAREZ, SUSAN C TROCK, MARC-ALAIN WIDDOWSON, DYLAN B GEORGE, MARC LIPSITCH AND JESSE D BLOOM

*For correspondence:
car44@cam.ac.uk

Influenza pandemics arise when antigenically novel influenza viruses enter and spread extensively in the human population. By this definition, there have been five influenza pandemics in the last 100 years, the worst of which cost 50 million lives worldwide (*Johnson and Mueller, 2002*). Of these pandemics, three likely arose from the introduction of genes from avian viruses into the human population (1918—H1N1, 1957—H2N2, 1968—H3N2 (*dos Reis et al., 2009*; *Neumann et al., 2009*, *Worobey et al., 2014*)), one arose from the introduction of a swine virus (2009—H1N1 (*Smith et al., 2009*)), and one was likely due to the unintended reintroduction of a previously widespread human virus that had not been seen in humans for two decades (1977—H1N1 (*dos Reis et al., 2009*, *Nakajima et al., 1978*, *Palese, 2004*)). However, the viruses responsible for these pandemics represent only a tiny fraction of the total diversity of influenza A viruses that exist in nature (*Webster et al., 1992*). Assessing

which viruses pose the greatest risk of causing the next human pandemic is an enormous challenge.

Pandemic influenza risk assessment faces a fundamental problem: a paucity of empirical data on the differences between pandemic viruses and their immediate ancestors from non-human hosts. The challenge was clearly articulated by Harvey Fineberg in his analysis of the US government's response to the 1976 swine influenza scare (*Fineberg, 2009*): 'The first lesson is to avoid over-confidence about scientific insights. Major flu pandemics arise on average only about three times every century, which means scientists can make relatively few direct observations in each lifetime and have a long time to think about each observation. That is a circumstance that is ripe for over-interpretation.'

Core elements of current approaches to pandemic preparedness and mitigation, such as the development of vaccines and stockpiling of antiviral drugs, require detailed virological and

immunological data on viruses with perceived pandemic potential and ample lead time for production (*Jennings et al., 2008*, *Keitel and Piedra, 2014*). The substantial diversity of known influenza viruses in non-human hosts, and the frequent identification of new viruses, makes extensive experimental testing and development of pandemic preparedness measures against all viruses unfeasible. Thus, there is a need for continuing attempts to assess the pandemic risks posed by non-human viruses in order to prioritize viruses of concern for pandemic preparedness planning. Currently, influenza pandemic risk assessment is largely driven by a simple idea: animal viruses that cause sporadic human infections are thought to pose a greater pandemic risk than viruses that have not been documented to infect humans (*Figure 1*). This intuitively attractive idea does not have direct empirical support, as none of the viruses that caused the 1918, 1957, 1968, or 2009 pandemics was detected in humans before they emerged in their pandemic form (*Smith et al., 2009*). This is largely due to a lack of surveillance (1918, 1957, and 1968 pandemics) and to the mistaken assumption that virus subtypes already circulating in humans were unlikely to cause pandemics (2009 pandemic) (*Peiris et al., 2012*). However, increased surveillance has probably improved the chance that the next pandemic virus will be identified prior to sustained human-to-human transmission.

If it is true that influenza surveillance has the possibility of identifying potential pandemic viruses before they begin to spread extensively between humans, then improving the basis for assessment of the risks posed by those viruses is an important goal. The level of public health concern about identified non-human influenza viruses should be a function of the potential of each virus to gain the ability to transmit efficiently from human to human and the severity of disease that such a virus would cause should it become pandemic. These two high-level phenotypes are each determined by the interaction of a number of biochemical traits of the virus during human infection (*Figure 2*) (*Chou et al., 2011*, *Hatta et al., 2001*, *Kobasa et al., 2004*, *Labadie et al., 2007*, *Yen et al., 2011*), the state of immunity to that influenza virus in human populations at the time of emergence (*Miller et al., 2010*, *Xu et al., 2010*), and by environmental factors such as temperature and humidity (*Shaman et al., 2011*).

Currently, the primary tool that uses multiple data streams for assessing pandemic risk is the Influenza Risk Assessment Tool (IRAT) (*Cox et al., 2014*, *Trock et al., 2012*). The IRAT integrates existing knowledge, including information on virus transmissibility and disease severity, with expert opinion about potential pandemic viruses to assign relative risk scores to those viruses. The IRAT is useful for identifying key gaps in knowledge,

| | Multiple human infections, high mortality rate (H5N1, H7N9) | Multiple human infections, low mortality rate (H3N2v) | Detect highly pathogenic avian virus* in a bird or mammal population | *In vivo* evidence for potential adaptation to mammals | *In vitro* evidence for potential adaptation to mammals | Computational genotype-to-phenotype predictions |
|---|---|---|---|---|---|---|
| Enhance surveillance | red | red | red | red | orange | yellow |
| Introduce animal control measures | red | orange | red | | | |
| Acquire seed strains for human vaccines | red | red | yellow | | | |
| Clinical trials and manufacture of pre-pandemic human vaccines | red | orange | | | | |
| Fill and finish non-adjuvanted human vaccines | red | orange | | | | |
| Fill and finish adjuvanted human vaccines | orange | yellow | | | | |

**Figure 1**. Evidence for concern and actions to mitigate influenza pandemics. Types of evidence that have been, or could be, used to justify specific preparedness or mitigation actions prior to evidence of sustained human-to-human transmission, largely based on the authors' interpretation of national and international responses to H5N1, H7N9, and H3N2v outbreaks (*Epperson et al., 2013*, *WHO, 2011*). Red indicates largely sufficient, orange partly sufficient, yellow minimally sufficient, gray insufficient. * high pathogenicity phenotype as defined by the World Organization for Animal Health (OIE) (*OIE, 2013*).

focusing risk management efforts, and providing clear documentation of decision rationales. However, to be used optimally, the IRAT requires a substantial amount of experimental data about virus phenotypes including information on receptor binding, transmissibility in laboratory animals, and antiviral treatment susceptibility. In the absence of phenotype data, preliminary assessments with the IRAT must rely on extrapolations from related viruses, which are prone to subjective interpretation.

The biochemical traits that determine virus phenotypes are themselves determined by the genetic sequence of the virus (*Figure 2*). In theory, it might eventually be possible to predict virus phenotype directly from virus sequence data. However, the complexities of the relationships between sequences and traits and from traits to disease phenotypes, make the prediction of pandemic potential from genomic sequence a tremendous challenge. Here, we discuss ways in which laboratory experiments, together with computational and theoretical developments, could improve genotype-to-phenotype prediction and, in conjunction with enhanced surveillance, improve assessment of the risks posed to humans by non-human influenza viruses.

## Experimental approaches

One goal of experimental studies on non-human influenza viruses is to identify general virus traits that are likely to affect transmissibility between humans, and then relate those traits to specific virus sequence changes. For obvious reasons, direct experimental assessment of human-to-human transmission of potential pandemic viruses is not feasible. However, influenza viruses that have caused pandemics in humans have been shown to transmit efficiently in animal models (most commonly ferrets) (*Chou et al., 2011*, *Yen et al., 2011*), thus animal models are thought to be useful for examining the genetic changes in viruses that facilitate human-to-human transmission. For example, several studies have shown that genetic changes in the neuraminidase (NA) and matrix (M) gene segments acquired by the virus lineage responsible for the 2009 H1N1 pandemic increased transmissibility in animal models (*Chou et al., 2011*, *Lakdawala et al., 2011*, *Yen et al., 2011*), suggesting that these changes may have played a role in enhancing the virus's transmissibility in humans and hence paved the way for pandemic emergence. When animal experiments provide quantitative measures of virus traits, these can be integrated into quantitative measures of risk assessment such as the IRAT (*Trock et al., 2012*).

Recently, several high-profile and controversial gain-of-function (GoF) studies have attempted to go beyond the characterization of existing viruses to prospectively identify new mutations in avian H5N1 viruses that enhance the ability of these viruses to transmit between ferrets by the airborne route (*Chen et al., 2012*, *Herfst et al., 2012*, *Imai et al., 2012*, *Zhang et al., 2013*). Important questions about the relative risks and benefits of these studies have been debated extensively elsewhere (*Fauci, 2012*; *Fouchier et al., 2013*; *Lipkin, 2012*; *Casadevall and Imperiale, 2014*; *Lipsitch and Galvani, 2014*); here, we focus on scientific considerations.

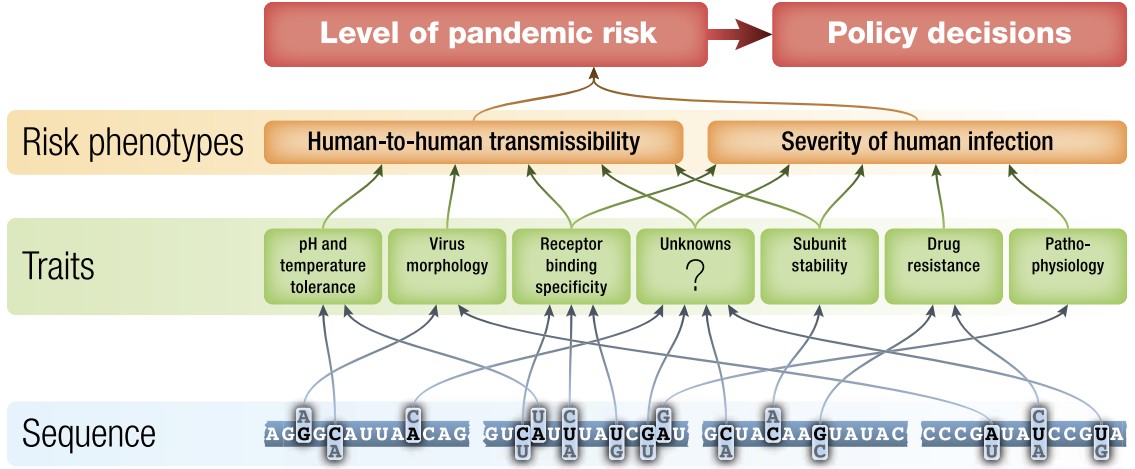

**Figure 2**. Schematic of potential relationships from virus genetic sequence to level of public health concern/pandemic risk. Pandemic risk is a combination of the probability that a virus will cause a pandemic and the human morbidity and mortality that might result from that pandemic. Arrows represent possible relationships between levels and are not intended to summarize current knowledge.

Because of the vast size of genetic space, such studies cannot possibly delineate all genetic variants of a virus that might be transmissible—after all, there are more than $10^{18}$ different possible five-mutation variants of any given hemagglutinin (HA), which is more than what can reasonably be assayed experimentally and the vast majority will not facilitate transmissibility. A more modest goal is to attempt to associate classes of genetic or phenotypic traits with transmissibility. Transmissibility traits identified by GoF studies to date include some that were already known (such as switching receptor binding from avian-like α2,3 sialic acid to human-like α2,6 sialic acid linkages (*Yamada et al., 2006*) and lowering the optimal temperature for viral polymerase activity (*Massin et al., 2001*)), as well as some that are new, such as increasing HA stability and reducing glycosylation on HA's globular head (*Herfst et al., 2012*, *Imai et al., 2012*). Whether these traits are either necessary or sufficient for transmissibility among humans or even other mammalian animal models remains unclear. For example, a recent study of an avian H5N1 virus found that by reassorting its internal genes with those of a 2009 pandemic virus, the virus could be rendered transmissible in guinea pigs (which have both α2,6 and α2,3 sialic acid in the upper respiratory tract) despite retaining a preference for binding α2,3 sialic acid. However, when mutations identified in earlier ferret GoF experiments were used to switch the receptor specificity to α2,6 sialic acid, transmissibility was lost (*Zhang et al., 2013*).

A key question for efforts to assess pandemic risk of non-human viruses is the degree to which certain substitutions are general markers for a phenotype, or whether the impacts of those mutations are dependent on genetic context and/or specific non-human host. Some mutations have been shown to be strong markers for phenotype for well-defined collections of viruses—for instance, the NA mutation H275Y consistently confers oseltamivir resistance on N1 neuraminidases (although the impact of the mutation on surface expression of NA, and thus virus fitness, varies dramatically) (*Baz et al., 2010*, *Bloom et al., 2010*). Similarly, the PB2 E627K substitution adapts the viral polymerase to mammalian cells in some viruses (*Long et al., 2013*) but not others (*Herfst et al., 2010*), while other viruses have adapted to mammals via different substitutions in PB2 (*Jagger et al., 2010*, *Mehle and Doudna, 2009*; *Zhu et al., 2010*). In many cases, the effect of mutations can be highly sensitive to genetic context—for instance, the effects of

cytotoxic T-lymphocyte escape mutations on nucleoprotein (NP) function depend on the stability of the parent protein, which can be affected by at least dozens of other mutations (*Gong et al., 2013*). Similar patterns of context dependence have recently been shown for receptor binding specificity substitutions in H5N1 viruses (*Tharakaraman et al., 2013*). Therefore, even when phenotypic traits of interest can be identified, clear genetic markers for these traits are only present in some cases.

The utility of experimental studies for informing surveillance for higher-risk viruses hinges on the question of whether virus traits associated with risk of infection and transmission in humans possess clear genetic markers. If a trait only arises from a limited number of specific mutations or combination of mutations, then experimentally delineating these mutations would be helpful for surveillance. For these cases, it is important and useful for the community to have access to collections of interpretable genotype to phenotype traits such as in the H5N1 genetic changes inventory (http://www.cdc.gov/flu/avianflu/h5n1-genetic-changes.htm) as well as computational tools to quickly connect new sequences to the body of available mutation annotation knowledge (FluSurver: http://flusurver.bii.a-star.edu.sg/). On the other hand, if a trait can be conferred by a large number of different mutations or combinations of mutations, then it will be less effective to monitor specific mutations. In such cases, it may be more beneficial to focus on the broader biochemical properties of viruses or their proteins. Developing laboratory capacity for rapid phenotype assessment would therefore be a valuable complement to high-throughput sequencing of new viruses. Moving forwards, if such biochemical traits can be clearly delineated and reliably modeled, then computational simulation of proteins could be used to predict phenotype from sequence, even for sequences from viruses that have never been experimentally tested.

## Computational predictions

Computational methods present an attractive adjunct to experimental studies because they have higher throughput, have shorter turnaround times, are cheaper, and are safer than experimental work with whole virulent viruses. The main drawback of computational methods is the largely unknown accuracy of their predictions—a drawback that is exacerbated by the lack of an established framework for validating the accuracy of the numerous computational prediction methods that populate the literature.

The elements of influenza pandemic risk assessment that are most amenable to computational prediction are those that correspond to well-defined, quantifiable molecular-scale traits such as receptor-binding preference, antiviral susceptibility, antigenicity of HA and NA, and possibly T-cell epitopes. Higher-level phenotypes such as transmissibility, that integrate phenomena at a range of scales, are not yet sufficiently well understood to be reasonable targets for computational predictions. A variety of computational methods shows promise for genotype-to-phenotype prediction including molecular dynamics simulations that combine high- and low-fidelity models (*Amaro et al., 2009*) and statistical learning approaches that use protein structure, dynamics, and sequence data to predict the phenotypic consequences of mutation (*Kasson et al., 2009*). However, better prospective validation of these tools against experimental data, particularly for exploring context dependency of genetic changes, is essential before these tools can be reliably used for informing public health decisions or policy-making (*Figure 1*).

Making substantial progress in the development of computational tools and the assessment of their accuracy will require collaboration between experimental and computational scientists to produce consistent testing and validation data. One possible mechanism to spur cooperation would be a series of regular community assessment exercises similar to Critical Assessment of protein Structure Prediction (CASP) (*Moult et al., 2011*). In a CASP-like exercise, one or more experimental groups would generate quantitative phenotype data for a set of viruses, for example the relative binding of α2,3-sialoglycans and α2,6-sialoglycans, pH profile of viral activation, or sensitivity to oseltamivir, and challenge computational groups to predict that virus phenotype data from the genetic sequences of the viruses tested. The quantitative experimental data would be held under embargo while the exercise runs. Computational groups would complete predictions for these targets, the experimental data set would then be released, and a meeting would be held to assess the performance of different methods to define avenues for improvement.

Ideal experimental data sets for CASP-like exercises include thermophoretic or interferometric measurements of HA binding affinities to α2,3- and α2,6-sialoglycans (*Xiong et al., 2013*) and multi-method characterizations of viral pH activation shifts for sets of point mutants in HA (*Galloway et al., 2013*, *Thoennes et al., 2008*). Reliable computational prediction of biochemical traits from genetic data would be a major accomplishment. However, it should be recognized that further major developments, particularly computational prediction of total virus fitness in new hosts, would still be required for realizing the utility of computational tools in policymaking.

## Evolutionary theory and modeling

In addition to the genotype–phenotype relationship itself, there is a need for better understanding of the evolutionary mechanisms and pathways that allow adaptive mutations controlling host range to appear and rise in frequency. These mechanisms act in reservoir hosts, in intermediate hosts (if any), and in humans or other potential hosts; they also act at multiple scales, as viruses compete for replication within hosts and transmission between hosts (*Park et al., 2013*, *Russell et al., 2012*). Developing better phylodynamic model frameworks (*Grenfell et al., 2004*) for modeling virus host transfer and adaptation will require collaboration between theoreticians and experimentalists.

Specific goals would be to determine realistic parameters for mutation/selection processes (*Illingworth et al., 2014*) and virus population bottlenecks at transmission (*Wilker et al., 2013*) and to generate high-resolution data sets to test and train mechanistic models. Such data-driven mechanistic models could shed light on additional constraints to virus genetic change, such as fitness valleys that separate virus genotypes adapted to one species or another, or conflicts in selection acting at different biological scales. For example, at the most simple level of understanding of the role of receptor binding, avian to mammalian host switching is often assumed to only require a binary change in receptor specificity from α2,3 to α2,6 sialic acid and to be directly related to binding affinity. However, in addition to the α2,3 and α2,6 linkages, there is a tremendous variety in the structures of oligosaccharides displaying the sialic acids and in the structure of the sialic acids in different avian hosts (*Gambaryan et al., 2012*, *Jourdain et al., 2011*). The binding specificity for each receptor variant form may affect the potential for different viruses to cross the species barrier or make the difference between causing severe or only mild disease. Rich experimental data sets that provide insights on such factors will improve the power of evolutionary models to interpret experimental and field data.

## Surveillance methodology

Detection of the genetic changes and phenotypes of concern relies on systematic characterization

of influenza viruses circulating in wild and domesticated animal populations. If there are virus traits that correlate with genetic markers observed to increase risk in humans, or that can be computationally inferred from genetic sequence data, it could be possible to monitor those markers in surveillance and adjust risk assessments prior to emergence in humans. However, the acquisition of samples entering existing surveillance networks is largely ad hoc, exhibits substantial variation by host and geographical region, and only a small proportion of the data end up in the public domain (*Figure 3*). Making non-human influenza surveillance more systematic by using statistical analysis to determine appropriate levels of coverage by geographic region and host species would facilitate the early detection of viruses of concern and also have the potential to facilitate detection of evolutionary and epidemiological patterns of virus activity that warn of potential emergence events.

There are large regions of the world and many animal populations for which little or no surveillance is performed but where significant

animal influenza diversity can be inferred to exist. Systematic assessments of surveillance by geographic area and host species, similar to efforts for malaria (*Gething et al., 2012*, *2011*, *Hay et al., 2010*, *Sinka et al., 2012*) and dengue (*Bhatt et al., 2013*), would help to identify major gaps where surveillance is either non-existent or unlikely to be sufficient for timely detection of viruses of concern. For enhancing surveillance, prioritizing among these gaps will require substantial improvements in understanding animal host ecology to identify hotspots for virus transmission within and among animal species. Similar efforts are required to better understand what aspects of the human–animal interface facilitate transmission of viruses between animals and humans, particularly in animal production and domestic animal settings, and the human biological and epidemiological factors that promote chains of transmission of newly introduced viruses.

One motivation for changing existing surveillance systems is to increase their power to rapidly detect changes in patterns of non-human influenza virus activity. Substantial changes, such as

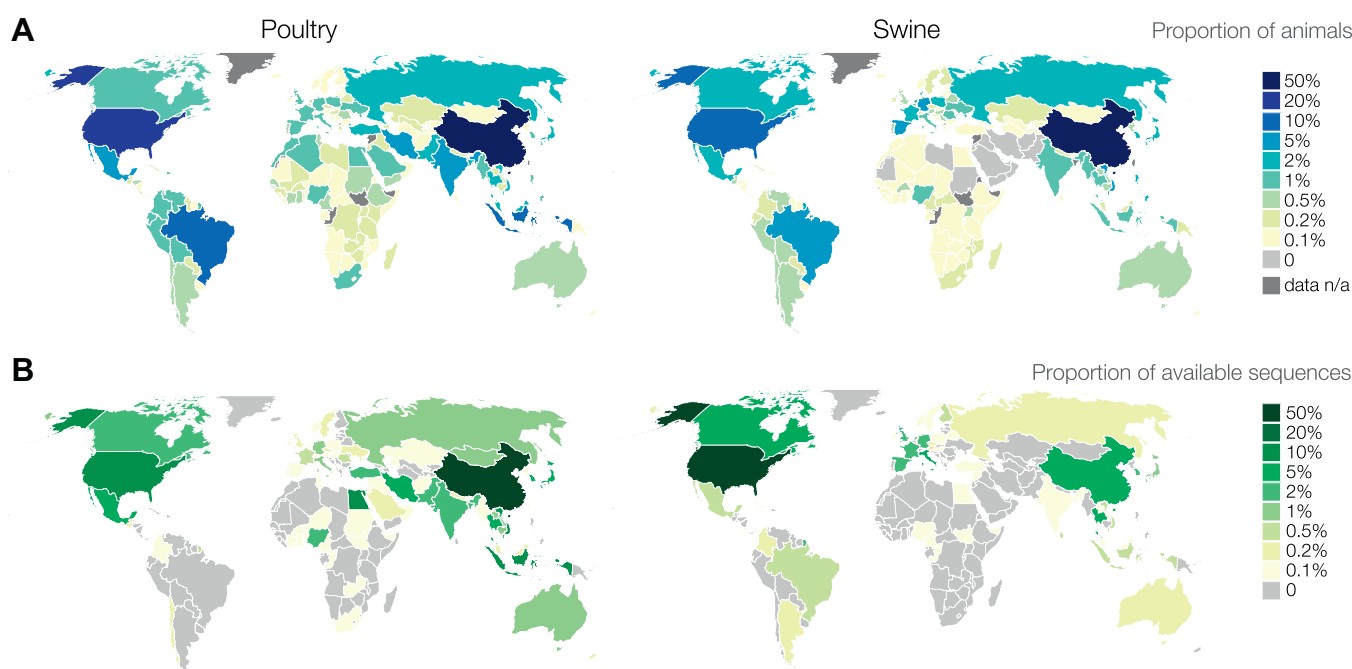

**Figure 3**. Geographic distribution of publicly available influenza virus genetic sequence data in comparison to poultry and swine populations. (**A**) Proportions of worldwide animal population by country (data from the Food and Agriculture Organization of the United Nations). (**B**) Number of unique influenza viruses for which sequence data exists in public databases from poultry or swine by country. Numbers of influenza virus sequences are not representative of influenza virus surveillance activities. Information regarding surveillance activities is not readily available. Virological surveillance, even if robust, may result in negative findings and is not captured in these figures. Most countries do not sequence every influenza virus isolate and some countries conduct virological surveillance without sharing sequence data publicly. Sequences deposited in public databases can reflect uneven geographic distribution and interest regarding viruses of concern such as H5N1 and H9N2.

the sudden proliferation of a previously rare virus subtype or of a virus with an H9N2 internal gene cassette (*Gao et al., 2013*, *GarcÌa-Sastre and Schmolke, 2014*; *Guan et al., 1999*), could indicate the emergence of new viral variants in non-human hosts that should be prioritized for further study even before the detection of human infections of zoonotic origin (*Vijaykrishna et al., 2011*). To be useful from a human health perspective such detection systems would require sampling of animals with no obvious signs of infection, routine assessment of particular genetic signatures or full genome sequencing, and near real-time sharing of these data; these activities all present potential financial, political, and logistical constraints.

Further development of surveillance infrastructure in some geographic locations and host species is likely to be unpopular or unfeasible due to economic disincentives for disease detection. However, the geographic movements of many non-human influenza hosts, via migration or trade, make it possible to identify surrogate sources of information. For example, by linking virological and serological data, it has been possible to make inferences about swine influenza virus activity in some parts of mainland China based only on the data from Hong Kong (*Strelioff et al., 2013*).

A systematic, open, and timely global surveillance system based on viral sequence data would be a powerful tool in pandemic risk assessment. Viral sequences, with associated metadata and systematic recording of virus negative sample results, provide a rich source of information beyond the simple presence or absence of particular strains. Phylodynamic reconstructions from even a relatively small number of samples are capable of revealing lineages that are proliferating (*Grenfell et al., 2004*, *Pybus and Rambaut, 2009*). Phylogenetic methods can be used to reveal gaps in surveillance (*Smith et al., 2009*, *Vijaykrishna et al., 2011*). Genetic similarity between viruses in different locations or host species can identify drivers of transmission between populations (*Faria et al., 2013*, *Lemey et al., 2014*).

Data on negative samples would provide valuable denominators for estimating the prevalence of infection: tracking infection rates through time would give a window into transmission dynamics and allow investigation of mechanisms underlying virus circulation. The Influenza Research Database (IRD) (http://www.fludb.org) includes an animal surveillance database that contains negative test data but the amount of data is extremely limited compared to the global scale of ongoing surveillance activities. Standards should be developed for consistently recording these relevant associated metadata, so that the number of animals tested, the setting in which sampling took place, and the motivation for sampling associated with genetic data can be submitted in a consistent form to public data repositories, along with all sequence submissions.

## Conclusions

It is currently not possible to predict which non-human influenza A virus will cause the next pandemic. Reducing the impact of the next pandemic will rely on early detection and mitigation strategies that slow the early spread to allow more preparatory work to be done. The integration of further experimental data with computational methods and mathematical models in conjunction with refinements to surveillance methodology will increase the feasibility of genotype-to-phenotype based assessments, increase the power of tools for more objectively assessing pandemic risk and decrease the time required for assessing the pandemic threat posed by extant non-human influenza A viruses—all of which can inform strategies to help mitigate the impact of the next pandemic.

Even as risk assessment capabilities improve, scientific insights into non-human influenza viruses must not give way to complacency that the most substantial threats have been identified and characterized. Despite the perceived risks of highly pathogenic H5N1 viruses, the emergence of the 2009 H1N1 pandemic virus in humans, the increasing incidence of human infection with H7N9 viruses in China since 2013, and the first documented human infections with H6N1 (*Wei et al., 2013*) and H10N8 (*Chen et al., 2014*) viruses highlight the importance of remaining vigilant against as-yet unrecognized high-risk viruses and the value of surveillance for influenza viruses in humans. Beyond further scientific investigations and refinement of surveillance capacity, the development of local surveillance-based outbreak response capacity worldwide remains essential. The first wave of the 2013 H7N9 outbreak in China demonstrated the value of swift coordinated action, including the timely dissemination of surveillance data, to limit further incursions of new viruses into the human population. Without developing similar response capacities in other areas at high risk of new virus introductions, we are only building expensive systems for watching the next pandemic unfold.

### Acknowledgements

This work was supported in part by the Research and Policy for Infectious Disease Dynamics (RAPIDD) program of the Science and Technology Directorate, U.S Department of Homeland Security,

and the Fogarty International Center, NIH. The findings and conclusions in this report are those of the authors and do not necessarily reflect the views of the Centers for Disease Control and Prevention, the Department of Health and Human Services or its components, or the United States Department of Agriculture.

**Colin A Russell** Department of Veterinary Medicine, University of Cambridge, Cambridge, United Kingdom

**Peter M Kasson** Department of Biomedical Engineering, University of Virginia, Charlottesville, United States

**Ruben O Donis** Influenza Division, Centers for Disease Control and Prevention, Atlanta, United States

**Steven Riley** Department of Infectious Disease Epidemiology, School of Public Health, Imperial College London, London, United Kingdom; Fogarty International Center, National Institutes of Health, Bethesda, United States

**John Dunbar** Bioscience Division, Los Alamos National Laboratory, Los Alamos, United States

**Andrew Rambaut** Fogarty International Center, National Institutes of Health, Bethesda, United States; Institute of Evolutionary Biology, University of Edinburgh, Edinburgh, United Kingdom

**Jason Asher** Leidos contract support to the Division of Analytic Decision Support, Biomedical Advanced Research and Development Authority, Department of Health and Human Services, Washington, United States

**Stephen Burke** Influenza Division, Centers for Disease Control and Prevention, Atlanta, United States

**C Todd Davis** Influenza Division, Centers for Disease Control and Prevention, Atlanta, United States

**Rebecca J Garten** Influenza Division, Centers for Disease Control and Prevention, Atlanta, United States

**Sandrasegaram Gnanakaran** Bioscience Division, Los Alamos National Laboratory, Los Alamos, United States

**Simon I Hay** Spatial Ecology and Epidemiology Group, Department of Zoology, University of Oxford, Oxford, United Kingdom
http://orcid.org/0000-0002-0611-7272

**Sander Herfst** Department of Viroscience, Postgraduate School of Molecular Medicine, Erasmus Medical Center, Rotterdam, Netherlands

**Nicola S Lewis** Department of Zoology, University of Cambridge, Cambridge, United Kingdom

**James O Lloyd-Smith** Fogarty International Center, National Institutes of Health, Bethesda, United States; Department of Ecology and Evolutionary Biology, University of California, Los Angeles, Los Angeles, United States

**Catherine A Macken** Bioscience Division, Los Alamos National Laboratory, Los Alamos, United States

**Sebastian Maurer-Stroh** Bioinformatics Institute, Agency for Science Technology and Research, Singapore, Singapore; National Public Health Laboratory, Communicable Diseases Division, Ministry of Health, Singapore, Singapore; School of Biological Sciences, Nanyang Technological University, Singapore, Singapore

**Elizabeth Neuhaus** Influenza Division, Centers for Disease Control and Prevention, Atlanta, United States

**Colin R Parrish** James A Baker Institute, College of Veterinary Medicine, Cornell University, Ithaca, United States

**Kim M Pepin** Fogarty International Center, National Institutes of Health, Bethesda, United States; National Wildlife Research Center, United States Department of Agriculture, Fort Collins, United States

**Samuel S Shepard** Influenza Division, Centers for Disease Control and Prevention, Atlanta, United States

**David L Smith** Fogarty International Center, National Institutes of Health, Bethesda, United States; Spatial Ecology and Epidemiology Group, Department of Zoology, University of Oxford, Oxford, United Kingdom; Sanaria Institute for Global Health and Tropical Medicine, Rockville, United States
http://orcid.org/0000-0003-4367-3849

**David L Suarez** Exotic and Emerging Avian Viral Diseases Research Unit, Southeast Poultry Research Laboratories, United States Department of Agriculture, Athens, United States

**Susan C Trock** Influenza Division, Centers for Disease Control and Prevention, Atlanta, United States

**Marc-Alain Widdowson** Influenza Division, Centers for Disease Control and Prevention, Atlanta, United States

**Dylan B George** Fogarty International Center, National Institutes of Health, Bethesda, United States; Division of Analytic Decision Support, Biomedical Advanced Research and Development Authority, Assistant Secretary for Preparedness and Response, Department of Health and Human Services, Washington, DC, United States

**Marc Lipsitch** Center for Communicable Disease Dynamics, Department of Epidemiology, Harvard School of Public Health, Boston, United States; Department of Immunology and Infectious Diseases, Harvard School of Public Health, Boston, United States
http://orcid.org/0000-0003-1504-9213

**Jesse D Bloom** Division of Basic Sciences, Fred Hutchinson Cancer Research Center, Seattle, United States
http://orcid.org/0000-0003-1267-3408

## Author contributions

CAR, PMK, ROD, SR, JD, AR, JA, SB, CTD, RJG, SG, SIH, SH, NSL, JOL-S, CAM, SM-S, EN, CRP, KMP, SSS, DLS, DLS, SCT, M-AW, DBG, ML, JDB, Conception and design, Drafting or revising the article

**Competing interests:** SIH: Reviewing editor, *eLife*. The other authors declare that no competing interests exist.

## Funding

| Funder | Grant reference number | Author |
| --- | --- | --- |
| U.S. Department of Homeland Security | Research and Policy for Infectious Disease Dynamics | Colin A Russell, Peter M Kasson, Ruben O Donis, Steven Riley, John Dunbar, Andrew Rambaut, Jason Asher, Stephen Burke, C Todd Davis, Rebecca J Garten, Sandrasegaram Gnanakaran, Simon I Hay, Sander Herfst, Nicola S Lewis, James O Lloyd-Smith, Catherine A Macken, Sebastian Maurer-Stroh, Elizabeth Neuhaus, Colin R Parrish, Kim M Pepin, Samuel S Shepard, David L Smith, David L Suarez, Susan C Trock, Marc-Alain Widdowson, Dylan B George, Marc Lipsitch, Jesse D Bloom |
| Fogarty International Center | Research and Policy for Infectious Disease Dynamics | Colin A Russell, Peter M Kasson, Ruben O Donis, Steven Riley, John Dunbar, Andrew Rambaut, Jason Asher, Stephen Burke, C Todd Davis, Rebecca J Garten, Sandrasegaram Gnanakaran, Simon I Hay, Sander Herfst, Nicola S Lewis, James O Lloyd-Smith, Catherine A Macken, Sebastian Maurer-Stroh, Elizabeth Neuhaus, Colin R Parrish, Kim M Pepin, Samuel S Shepard, David L Smith, David L Suarez, Susan C Trock, Marc-Alain Widdowson, Dylan B George, Marc Lipsitch, Jesse D Bloom |

The funders had no role in study design, data collection and interpretation, or the decision to submit the work for publication.

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
