## [Decision Letter]

Thank you for sending your work entitled “Improving pandemic influenza risk assessment” for consideration at eLife. Your article has been favorably evaluated by Prabhat Jha (Senior editor), Sema Sgaier (Reviewing editor), and two peer reviewers. There are only a few minor comments from the reviewers that we would like for you to address before publication.

1) Author contributions: there are currently 25+ authors listed. One suggestion for you to consider is listing a few as main others and others as a group.

2) Provide more references for the statements made throughout the paper; there are many statements where references would be helpful. Please review the text and address this where possible.

3) Figure 2: Please change the label “Level of overall concern” to “Level of public health concern/pandemic risk” to be consistent with the figure legend.

4) Abstract suggested change: “it should become possible to predict pandemic potential from sequence data.”

---

## [Author Response]

*1) Author contributions: there are currently 25+ authors listed. One suggestion for you to consider is listing a few as main others and others as a group*.

This paper was born out of a workshop-based meeting on assessing the pandemic risk of influenza viruses. The level of interaction and discussion at the meeting was truly fantastic and it was the quality of the meeting that led us to writing the paper. Everyone at the meeting genuinely contributed to the production of the paper; there are no honorary authors here.

While we appreciate that the author list is somewhat lengthy, 28 authors to be exact, the group that produced this paper comprises opinion leaders in basic science and public health from diverse backgrounds. We feel that consensus from such a diverse group makes a powerful statement and would like to keep the author list as originally submitted.

*2) Provide more references for the statements made throughout the paper; there are many statements where references would be helpful. Please review the text and address this where possible*.

We have carefully gone through the text and added 20 new references.

*3)*
Figure 2*: Please change the label “Level of overall concern” to “Level of public health concern/pandemic risk” to be consistent with the figure legend*.

Due to the length of the phrase “Level of public health concern/pandemic risk”, making this change to Figure 2 would require restructuring the figure. We have changed the label to read “Level of pandemic risk”, which we feel links well with the figure caption.

*4) Abstract suggested change: “it should become possible to predict pandemic potential from sequence data*.*”*

We think the current wording, though longer, better reflects our thinking. We do not know if or when it will be possible to predict the total pandemic risk of a particular virus from sequence data – more work is needed first. In this sentence we want to speak to the potential utility of such predictions and push them as a direction for exploration.